# Learning metrics for persistence-based summaries and applications for graph classification

**Qi Zhao**
zhao.2017@osu.edu

**Yusu Wang**
yusu@cse.ohio-state.edu

Computer Science and Engineering Department
The Ohio State University
Columbus, OH 43221

## Abstract

Recently a new feature representation framework based on a topological tool called persistent homology (and its persistence diagram summary) has gained much momentum. A series of methods have been developed to map a persistence diagram to a vector representation so as to facilitate the downstream use of machine learning tools. In these approaches, the importance (weight) of different persistence features are usually *pre-set*. However often in practice, the choice of the weight-function should depend on the nature of the specific data at hand. It is thus highly desirable to *learn* a best weight-function (and thus metric for persistence diagrams) from labelled data. We study this problem and develop a new weighted kernel, called *WKPI*, for persistence summaries, as well as an optimization framework to learn the weight (and thus kernel). We apply the learned kernel to the challenging task of graph classification, and show that our WKPI-based classification framework obtains similar or (sometimes significantly) better results than **the best results** from a range of previous graph classification frameworks on benchmark datasets.

## 1 Introduction

In recent years a new data analysis methodology based on a topological tool called persistent homology has started to attract momentum. The persistent homology is one of the most important developments in the field of topological data analysis, and there have been fundamental developments both on the theoretical front (e.g, [23, 10, 13, 8, 14, 5]), and on algorithms / implementations (e.g, [43, 4, 15, 20, 29, 3]). On the high level, given a domain $X$ with a function $f : X \to \mathbb{R}$ on it, the persistent homology summarizes "features" of $X$ across multiple scales simultaneously in a single summary called the *persistence diagram* (see the second picture in Figure 1). A persistence diagram consists of a multiset of points in the plane, where each point $p = (b, d)$ intuitively corresponds to the birth-time ($b$) and death-time ($d$) of some (topological) features of $X$ w.r.t. $f$. Hence it provides a concise representation of $X$, capturing *multi-scale features* of it simultaneously. Furthermore, the persistent homology framework can be applied to complex data (e.g, 3D shapes, or graphs), and different summaries could be constructed by putting different descriptor functions on input data.

Due to these reasons, a new persistence-based feature vectorization and data analysis framework (Figure 1) has become popular. Specifically, given a collection of objects, say a set of graphs modeling chemical compounds, one can first convert each shape to a persistence-based representation. The input data can now be viewed as a set of points in a persistence-based feature space. Equipping this space with appropriate distance or kernel, one can then perform downstream data analysis tasks (e.g, clustering).

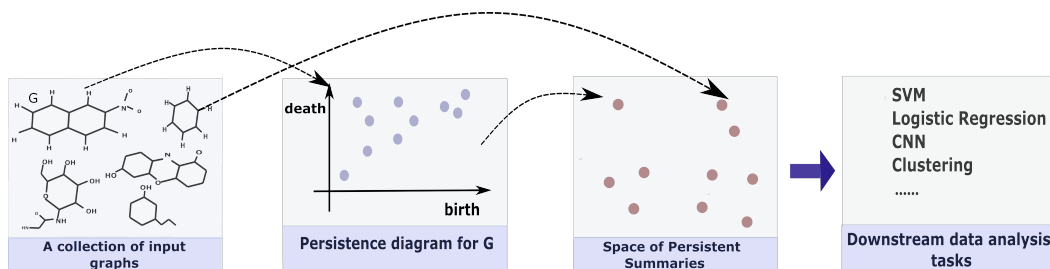

Figure 1: A persistence-based data analysis framework.

The original distances for persistence diagram summaries unfortunately do not lend themselves easily to machine learning tasks. Hence in the last few years, starting from the persistence landscape [7], there have been a series of methods developed to map a persistence diagram to a vector representation to facilitate machine learning tools [41, 1, 33, 12, 35]. Recent ones include Persistence Scale-Space kernel [41], Persistence Images [1], Persistence Weighted Gaussian kernel (PWGK) [33], Sliced Wasserstein kernel [12], and Persistence Fisher kernel [34].

In these approaches, when computing the distance or kernel between persistence summaries, the importance (weight) of different persistence features are often *pre-determined*. In persistence images [1] and PWGK [33], the importance of having a **weight-function** for the birth-death plane (containing the persistence points) has been emphasized and explicitly included in the formulation of their kernels. However, before using these kernels, the weight-function needs to be pre-set.

On the other hand, as recognized by [26], the choice of the weight-function should depend on the nature of the specific type of data at hand. For example, for the persistence diagrams computed from atomic configurations of molecules, features with small persistence could capture the local packing patterns which are of utmost importance and thus should be given a larger weight; while in many other scenarios, small persistence leads to noise with low importance. However, in general researchers performing data analysis tasks may not have such prior insights on input data. Thus it is natural and highly desirable to *learn* a best weight-function from labelled data.

**Our work.**    We study the problem of learning an appropriate metric (kernel) for persistence summaries from labelled data, and apply the learnt kernel to the challenging graph classification task.

*(1) Metric learning for persistence summaries:*  We propose a new weighted-kernel (called *WKPI*), for persistence summaries based on persistence images representations. Our WKPI kernel is positive semi-definite and its induced distance is stable. A weight-function used in this kernel directly encodes the importance of different locations in the persistence diagram. We next model the metric learning problem for persistence summaries as the problem of learning (the parameters of) this *weight-function* from a certain function class. In particular, the metric-learning is formulated as an optimization problem over a specific cost function we propose. This cost function has a simple matrix view which helps both conceptually clarify its meaning and simplify the implementation of its optimization.

*(2) Graph classification application:*  Given a set of objects with class labels, we first learn a best WKPI-kernel as described above, and then use the learned WKPI to further classify objects. We implemented this *WKPI-classification framework*, and apply it to a range of graph data sets. Graph classification is an important problem, and there has been a large literature on developing effective graph representations (e.g, [25, 40, 2, 32, 44, 47, 38], including the very recent persistent-homology enhanced WL-kernel [42]), and graph neural networks (e.g, graph neural networks [48, 39, 46, 45, 35, 31]) to classify graphs. The problem is challenging as graph data are less structured. We perform our WKPI-classification framework on various benchmark graph data sets as well as new neuron-cell data sets. Our learnt WKPI performs consistently better than other persistence-based kernels. Most importantly, when compared with existing state-of-the-art graph classification frameworks, our framework shows similar or (sometimes significantly) better performance in almost all cases than the *best results* by existing approaches.

We note that [26] is the first to recognize the importance of using labelled data to learn a task-optimal representation of topological signatures. They developed an end-to-end deep neural network for this purpose, using a novel and elegant design of the input layer to implicitly learn a task-specific

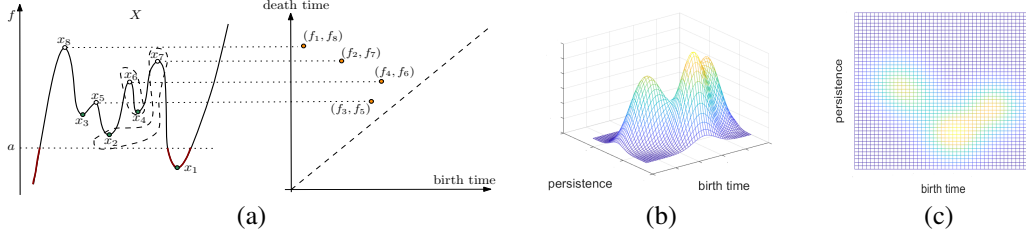

Figure 2: (a): As we sweep the curve bottom-up in increasing $f$-values, at certain critical moments new 0-th homological features (connected components) are created, or destroyed (i.e, components merge). For example, a component is created when passing $x_4$ and killed when passing $x_6$, giving rise to the persistence-point $(f_4, f_6)$ in the persistence diagram ($f_i := f(x_i)$). (b) shows the graph of a persistence surface (where $z$-axis is the function $\rho_A$), and (c) is its corresponding persistence image.

representation. Very recently, in a parallel and independent development of our work, Carrière et al. [11] built an interesting new neural network based on the DeepSet architecture [49], which can achieve an end-to-end learning for multiple persistence representations *in a unified manner*. Compared to these developments, we instead explicitly formulate the metric-learning problem for persistence-summaries, and decouple the metric-learning (which can also be viewed as representation-learning) component from the downstream data analysis tasks. Also as shown in Section 4, our WKPI-classification framework (using SVM) achieves better results on graph classification datasets.

## 2 Persistence-based framework

We first give an informal description of persistent homology below. See [22] for more detailed exposition on the subject.

Suppose we are given a shape $X$ (in our later graph classification application, $X$ is a graph). Imagine we inspect $X$ through a *filtration of $X$*, which is a sequence of growing subsets of $X$: $X_1 \subseteq X_2 \subseteq \cdots \subseteq X_n = X$. As we scan $X$, sometimes a new feature appears in $X_i$, and sometimes an existing feature disappears upon entering $X_j$. Using the topological object called homology classes to describe these features (intuitively components, independent loops, voids, and their high dimensional counter-parts), the birth and death of topological features can be captured by the *persistent homology*, in the form of a *persistence diagram* $\mathrm{Dg}X$. Specifically, for each dimension $k$, $\mathrm{Dg}_k X$ consists of a multi-set of points in the plane (which we call the *birth-death plane* $\mathbb{R}^2$): each point $(b, d)$ in it, called a *persistence-point*, indicates that a certain $k$-dimensional homological feature is created upon entering $X_b$ and destroyed upon entering $X_d$. In the remainder of the paper, we often omit the dimension $k$ for simplicity: when multiple dimensions are used for persistence features, we will apply our construction to each dimension and concatenate the resulting vector representations.

A common way to obtain a meaningful filtration of $X$ is via the *sublevel-set filtration* induced by a *descriptor function* $f$ on $X$. More specifically, given a function $f : X \to \mathbb{R}$, let $X_{\leq a} := \{x \in X \mid f(x) \leq a\}$ be its *sublevel-set at $a$*. Let $a_1 < a_2 < \cdots < a_n$ be $n$ real values. The sublevel-set filtration w.r.t. $f$ is: $X_{\leq a_1} \subseteq X_{\leq a_2} \subseteq \cdots \subseteq X_{\leq a_n}$; and its persistence diagram is denoted by $\mathrm{Dg}f$. Each persistence-point $p = (a_i, a_j) \in \mathrm{Dg}f$ indicates the function values when some topological features are created (when entering $X_{\leq a_i}$) and destroyed (in $X_{\leq a_j}$), and the *persistence* of this feature is its life-time $\mathrm{pers}(p) = |a_j - a_i|$. See Figure 2 (a) for a simple example where $X = \mathbb{R}$. If one sweeps $X$ top-down in decreasing function values, one gets the persistence diagram induced by the super-levelset filtration of $X$ w.r.t. $f$ in an analogous way. Finally, if one tracks the change of topological features in the *levelset* $f^{-1}(a)$, one obtains the so-called *levelset zigzag persistence* [9] (which contains the information captured by the *extended persistence* [17]).

**Graph Setting.** Given a graph $G = (V, E)$, a descriptor function $f$ defined on $V$ or $E$ will induce a filtration and its persistence diagrams. Suppose $f : V \to \mathbb{R}$ is defined on the node set of $G$ (e.g, the node degree). Then we can extend $f$ to $E$ by setting $f(u, v) = max\{f(u), f(v)\}$, and the sublevel-set at $a$ is defined as $G_{\leq a} := \{\sigma \in V \cup E \mid f(\sigma) \leq a\}$. Similarly, if we are given $f : E \to \mathbb{R}$, then we can extend $f$ to $V$ by setting $f(u) = \min_{u \in e, e \in E} f(e)$. When scanning $G$

via the sublevel-set filtration of $f$, connected components in the swept subgraphs will be created and merged, and new cycles will be created. The formal events are encoded in the 0-dimensional persistence diagram. The the 1-dimensional features (cycles), however, we note that cycles created will never be killed, as they are present in the total space $X = G$. To this end, we use the so-called *extended persistence* introduced in [17] which can record information of cycles.

Now given a collection of shapes $\Xi$, we can compute a persistence diagram $DgX$ for each $X \in \Xi$, which maps the set $\Xi$ to a set of points in the space of persistence diagrams. There are natural distances defined for persistence diagrams, including the bottleneck distance and the Wasserstein distance, both of which have been well studied (e.g, stability under them [16, 18, 14]) with efficient implementations available [27, 28]. However, to facilitate downstream machine learning tasks, it is desirable to further map the persistence diagrams to another "vector" representation. Below we introduce one such representation, called the persistence images [1], as our new kernel is based on it.

Let $A$ be a persistence diagram (containing a multiset of persistence-points). Following [1], set $T : \mathbb{R}^2 \to \mathbb{R}^2$ to be the linear transformation[1] where for each $(x, y) \in \mathbb{R}^2$, $T(x, y) = (x, y - x)$. Let $T(A)$ be the transformed diagram of $A$. Let $\phi_u : \mathbb{R}^2 \to \mathbb{R}$ be a differentiable probability distribution with mean $u \in \mathbb{R}^2$ (e.g, the normalized Gaussian where for any $z \in \mathbb{R}^2$, $\phi_u(z) = \frac{1}{2\pi\tau^2} e^{-\frac{\|z-u\|^2}{2\tau^2}}$).

**Definition 2.1 ([1])** *Let $\alpha : \mathbb{R}^2 \to \mathbb{R}$ be a non-negative weight-function for the persistence plane $\mathbb{R}^2$. Given a persistence diagram $A$, its* persistence surface *$\rho_A : \mathbb{R}^2 \to \mathbb{R}$ (w.r.t. $\alpha$) is defined as: for any $z \in \mathbb{R}^2$, $\rho_A(z) = \sum_{u \in T(A)} \alpha(u)\phi_u(z)$.*

*The persistence image is a discretization of the persistence surface. Specifically, fix a grid on a rectangular region in the plane with a collection $\mathcal{P}$ of $N$ rectangles (pixels). The* persistence image *for a diagram $A$ is $PI_A = \{ PI[\mathbf{p}] \}_{\mathbf{p} \in \mathcal{P}}$ consists of $N$ numbers (i.e, a vector in $\mathbb{R}^N$), one for each pixel $\mathbf{p}$ in the grid $\mathcal{P}$ with $PI[\mathbf{p}] := \iint_{\mathbf{p}} \rho_A \, dydx$.*

The persistence image can be viewed as a vector in $\mathbb{R}^N$. One can then compute distance between two persistence diagrams $A_1$ and $A_2$ by the $L_2$-distance $\|PI_1 - PI_2\|_2$ between their persistence images (vectors) $PI_1$ and $PI_2$. The persistence images have several nice properties, including stability guarantees; see [1] for more details.

## 3   Metric learning frameworks

Suppose we are given a set of $n$ objects $\Xi$ (sampled from a hidden data space $\mathcal{S}$), classified into $k$ classes. We want to use these labelled data to learn a good distance for (persistence image representations of) objects from $\Xi$ which hopefully is more appropriate at classifying objects in the data space $\mathcal{S}$. To do so, below we propose a new persistence-based kernel for persistence images, and then formulate an optimization problem to learn the best weight-function so as to obtain a good distance metric for $\Xi$ (and data space $\mathcal{S}$).

### 3.1   Weighted persistence image kernel (WKPI)

From now on, we fix the grid $\mathcal{P}$ (of size $N$) to generate persistence images (so a persistence image is a vector in $\mathbb{R}^N$). Let $p_s$ be the center of the $s$-th pixel $\mathbf{p}_s$ in $\mathcal{P}$, for $s \in \{1, 2, \cdots, N\}$. We now propose a new kernel for persistence images. A *weight-function* refers to a non-negative real-valued function on $\mathbb{R}^2$.

**Definition 3.1** *Let* $\omega : \mathbb{R}^2 \to \mathbb{R}$ *be a weight-function. Given two persistence images* $\mathrm{PI}$ *and* $\mathrm{PI}'$*, the* $(\omega\text{-})$*weighted persistence image kernel (WKPI) is defined as:* $k_w(\mathrm{PI}, \mathrm{PI}') :=$ $\sum_{s=1}^{N} \omega(p_s) e^{-\frac{(\mathrm{PI}(s) - \mathrm{PI}'(s))^2}{2\sigma^2}}$ .

*Remark 0:* We could use the persistence surfaces (instead of persistence images) to define the kernel (with the summation replaced by an integral). Since for computational purpose, one still needs to approximate the integral in the kernel via some discretization, we choose to present our work using persistence images directly. Our Lemma 3.2 and Theorem 3.4 still hold (with slightly different stability bound) if we use the kernel defined for persistence surfaces.

*Remark 1:* One can choose the weight-function from different function classes. Two popular choices are: mixture of $m$ 2D Gaussians; and degree-$d$ polynomials on two variables.

*Remark 2:* There are other natural choices for defining a weighted kernel for persistence images. For example, we could use $k(\mathrm{PI}, \mathrm{PI}') = \sum_{s=1}^{N} e^{-\frac{\omega(p_s)(\mathrm{PI}(s) - \mathrm{PI}'(s))^2}{2\sigma^2}}$ , which we refer this as *altWKPI*. Alternatively, one could use the weight function used in PWGK kernel [33] directly. Indeed, we have implemented all these choices, and our experiments show that our WKPI kernel leads to better results than these choices for almost all datasets (see Supplement Section 2). In addition, note that PWGK kernel [33] contains cross terms $\omega(x) \cdot \omega(y)$ in its formulation, meaning that there are quadratic number of terms (w.r.t the number of persistence points) to calculate the kernel, making it more expensive to compute and learn for complex objects (e.g, for the neuron data set, a single neuron tree could produce a persistence diagrams with hundreds of persistence points).

**Lemma 3.2** *The WKPI kernel is positive semi-definite.*

The rather simple proof of the above lemma is in Supplement Section 1.1. By Lemma 3.2, the WKPI kernel gives rise to a Hilbert space. We can now introduce the WKPI-distance, which is the *pseudo-metric* induced by the inner product on this Hilbert space.

**Definition 3.3** *Given two persistence diagrams $A$ and $B$, let $\mathrm{PI}_A$ and $\mathrm{PI}_B$ be their corresponding persistence images. Given a weight-function $\omega : \mathbb{R}^2 \to \mathbb{R}$, the $(\omega\text{-weighted})$ WKPI-distance is:*

$$\mathrm{D}_\omega(A, B) := \sqrt{k_w(\mathrm{PI}_A, \mathrm{PI}_A) + k_w(\mathrm{PI}_B, \mathrm{PI}_B) - 2k_w(\mathrm{PI}_A, \mathrm{PI}_B)}.$$

**Stability of WKPI-distance.** Given two persistence diagrams $A$ and $B$, two traditional distances between them are the bottleneck distance $d_B(A, B)$ and the $p$-th Wasserstein distance $d_{W,p}(A, B)$. Stability of these two distances w.r.t. changes of input objects or functions defined on them have been studied [16, 18, 14]. Similar to the stability study on persistence images, below we prove WKPI-distance is stable w.r.t. small perturbation in persistence diagrams as measured by $d_{W,1}$. (Very informally, view two persistence diagrams $A$ and $B$ as two (appropriate) measures (with special care taken to the diagonals), and $d_{W,1}(A, B)$ is roughly the "earth-mover" distance between them to convert the measure corresponding to $A$ to that for $B$.)

To simplify the presentation of Theorem 3.4, we use *unweighted persistence images w.r.t. Gaussian*, meaning in Definition 2.1, (1) the weight function $\alpha$ is the constant function $\alpha = 1$; and (2) the distribution $\phi_u$ is the Gaussian $\phi_u(z) = \frac{1}{2\pi\tau^2} e^{-\frac{\|z-u\|^2}{2\tau^2}}$ . (Our result below can be extended to the case where $\phi_u$ is not Gaussian.) The proof of the theorem below follows from results of [1] and can be found in Supplement Section 1.2.

**Theorem 3.4** *Given a weight-function $\omega : \mathbb{R}^2 \to \mathbb{R}$, set $c_w = \|\omega\|_\infty = \sup_{z \in \mathbb{R}^2} \omega(z)$. Given two persistence diagrams $A$ and $B$, with corresponding persistence images $\mathrm{PI}_A$ and $\mathrm{PI}_B$, we have that:* $\mathrm{D}_\omega(A, B) \le \sqrt{\frac{20c_w}{\pi}} \cdot \frac{1}{\sigma \cdot \tau} \cdot d_{W,1}(A, B)$, *where $\sigma$ is the width of the Gaussian used to define our WKPI kernel (Def. 3.1), and $\tau$ is that for the Gaussian $\phi_u$ to define persistence images (Def. 2.1).*

*Remarks:* We can obtain a more general bound for the case where the distribution $\phi_u$ is not Gaussian. Furthermore, we can obtain a similar bound when our WKPI-kernel and its induced WKPI-distance is defined using *persistence surfaces* instead of *persistence images*.

## 3.2 Optimization problem for metric-learning

Suppose we are given a collection of objects $\Xi = \{X_1, \ldots, X_n\}$ (sampled from some hidden data space $\mathcal{S}$), already classified (labeled) to $k$ classes $\mathcal{C}_1, \ldots, \mathcal{C}_k$. In what follows, we say that $i \in \mathcal{C}_j$ if $X_i$ has class-label $j$. We first compute the persistence diagram $A_i$ for each object $X_i \in \Xi$. (The precise filtration we use to do so will depend on the specific type of objects. Later in Section 4, we will describe filtrations used for graph data). Let $\{A_1, \ldots, A_n\}$ be the resulting set of persistence diagrams. Given a weight-function $\omega$, its induced WKPI-distance between $A_i$ and $A_j$ can also be thought of as a distance for the original objects $X_i$ and $X_j$; that is, we can set $D_\omega(X_i, X_j) := D_\omega(A_i, A_j)$. Our goal is to learn a good distance metric for the data space $\mathcal{S}$ (where $\Xi$ are sampled from) from the labels. We will formulate this as learning a best weight-function $\omega^*$ so that its induced WKPI-distance fits the class-labels of $X_i$'s best. Specifically, for any $t \in \{1, 2, \cdots, k\}$, set:

$$cost_\omega(t, t) = \sum_{i,j \in \mathcal{C}_t} D_\omega{}^2(A_i, A_j); \quad \text{and} \quad cost_\omega(t, \cdot) = \sum_{i \in \mathcal{C}_t, j \in \{1, 2, \cdots, n\}} D_\omega{}^2(A_i, A_j).$$

Intuitively, $cost_\omega(t, t)$ is the total in-class (square) distances for $\mathcal{C}_t$; while $cost_\omega(t, \cdot)$ is the total distance from objects in class $\mathcal{C}_t$ to all objects in $\Xi$. A good metric should lead to relatively smaller distance between objects from the same class, but larger distance between objects from different classes. We thus propose the following optimization problem, which is related to $k$-way spectral clustering where the distance for an edge $(A_i, A_j)$ is $D_\omega^2(A_i, A_j)$:

**Definition 3.5 (Optimization problem)** *Given a weight-function $\omega : \mathbb{R}^2 \to \mathbb{R}$, the* total-cost *of its induced WKPI-distance over $\Xi$ is defined as: $TC(\omega) := \sum_{t=1}^k \frac{cost(t,t)}{cost(t,\cdot)}$. The* optimal distance problem *aims to find the best weight-function $\omega^*$ from a certain function class $\mathcal{F}$ so that the total-cost is minimized; that is: $TC^* = \min_{\omega \in \mathcal{F}} TC(\omega)$; and $\omega^* = \operatorname{argmin}_{\omega \in \mathcal{F}} TC(\omega)$.*

**Matrix view of optimization problem.** We observe that our cost function can be re-formulated into a matrix form. This provides us with a perspective from the Laplacian matrix of certain graphs to understand the cost function, and helps to simplify the implementation of our optimization problem, as several programming languages popular in machine learning (e.g Python and Matlab) handle matrix operations more efficiently (than using loops). More precisely, recall our input is a set $\Xi$ of $n$ objects with labels from $k$ classes. We set up the following matrices:

$$L = G - \Lambda; \quad \Lambda = \big[\Lambda_{ij}\big]_{n \times n}, \quad \text{where } \Lambda_{ij} = D_\omega{}^2(A_i, A_j) \text{ for } i, j \in \{1, 2, \cdots, n\};$$

$$G = \big[g_{ij}\big]_{n \times n}, \quad \text{where } g_{ij} = \begin{cases} \sum_{\ell=1}^n \Lambda_{i\ell} & \text{if } i = j \\ 0 & \text{if } i \neq j \end{cases}$$

$$H = \big[h_{ti}\big]_{k \times n} \quad \text{where } h_{ti} = \begin{cases} \frac{1}{\sqrt{cost_\omega(t,\cdot)}} & i \in \mathcal{C}_t \\ 0 & otherwise \end{cases}$$

Viewing $\Lambda$ as distance matrix of objects $\{X_1, \ldots, X_n\}$, $L$ is then its Laplacian matrix. We have the following main theorem, which essentially is similar to the trace-minimization view of $k$-way spectral clustering (see e.g, Section 6.5 of [30]). The proof for our specific setting is in Supplement 1.3.

**Theorem 3.6** *The total-cost can also be represented by $TC(\omega) = k - \operatorname{Tr}(HLH^T)$, where $\operatorname{Tr}(\cdot)$ is the trace of a matrix. Furthermore, $HGH^T = \mathbf{I}$, where $\mathbf{I}$ is the $k \times k$ identity matrix.*

Note that all matrices, $L, G, \Lambda$, and $H$, are dependent on the (parameters of) weight-function $\omega$, and in the following corollary of Theorem 3.6, we use the subscript of $\omega$ to emphasize this dependence.

**Corollary 3.7** *The Optimal distance problem is equivalent to*

$$\min_\omega \big(k - \operatorname{Tr}(H_\omega L_\omega H_\omega^T)\big), \quad \text{subject to } H_\omega G_\omega H_\omega^T = \mathbf{I}.$$

**Solving the optimization problem.**   In our implementation, we use (stochastic) gradient descent to find a (locally) optimal weight-function $\omega^*$ for the minization problem. Specifically, given a collection of objects $\Xi$ with labels from $k$ classes, we first compute their persistence diagrams via appropriate filtrations, and obtain a resulting set of persistence diagrams $\{A_1, \ldots, A_n\}$. We then aim to find the best parameters for the weight-function $\omega^*$ to minimize $Tr(HLH^T) = \sum_{t=1}^{k} h_t L h_t^T$ subject to $HGH^T = I$ (via Corollary 3.7). For example, assume that the weight-function $\omega$ is from the class $\mathcal{F}$ of mixture of $m$ number of 2D non-negatively weighted (spherical) Gaussians. Each weight-function $\omega : \mathbb{R}^2 \to \mathbb{R} \in \mathcal{F}$ is thus determined by $4m$ parameters $\{x_r, y_r, \sigma_r, w_r \mid r \in \{1, 2, \cdots, m\}\}$ with $\omega(z) = w_r e^{-\frac{(z_x - x_r)^2 + (z_y - y_r)^2}{\sigma_r^2}}$ . We then use (stochastic) gradient decent to find the best parameters to minimize $Tr(HLH^T)$ subject to $HGH^T = I$. Note that the set of persistence diagrams / images will be fixed through the optimization process.

From the proof of Theorem 3.6 (in Supplement 1.3), it turns out that condition $HGH^T = \mathbf{I}$ is satisfied as long as the multiplicative weight $w_r$ of each Gaussian in the mixture is non-negative. Hence during the gradient descent, we only need to make sure that this holds [2]. It is easy to write out the gradient of $TC(\omega)$ w.r.t. each parameter $\{x_r, y_r, \sigma_r, w_r \mid r \in \{1, 2, \cdots, m\}\}$ **in matrix form**. For example, $\frac{\partial TC(\omega)}{\partial x_r} = -(\sum_{t=1}^{k} \frac{\partial h_t}{\partial x_r} L h_t^T + h_t \frac{\partial L}{\partial x_r} h_t^T + h_t L \frac{\partial h_t^T}{\partial x_r})$; where $h_t = \begin{bmatrix} h_{t1}, h_{t2}, ..., h_{tn} \end{bmatrix}$ is the $t$-th row vector of $H$. While this does not improve the asymptotic complexity of computing the gradient (compared to using the formulation of cost function in Definition 3.5), these matrix operations can be implemented much more efficiently than using loops in languages such as Python and Matlab. For large data sets, we use stochastic gradient decent, by sampling a subset of $s << n$ number of input persistence images, and compute the matrices $H, D, L, G$ as well as the cost using the subsampled data points. The time complexity of one iteration in updating parameters is $O(s^2 N)$, where $N$ is the size of a persistence image (recall, each persistence image is a vector in $\mathbb{R}^N$). In our implementation, we use Armijo-Goldstein line search scheme to update the parameters in each (stochastic) gradient decent step. The optimization procedure terminates when the cost function converges or the number of iterations exceeds a threshold. Overall, the time complexity of our optimization procedure is $O(\mathrm{R}s^2 N)$ where $\mathrm{R}$ is the number of iterations, $s$ is the minibatch size, and $N$ is the size (# pixels) of a single persistence image.

# 4   Experiments

We show the effectiveness of our metric-learning framework and the use of the learned metric via graph classification applications. In particular, given a set of graphs $\Xi = \{G_1, \ldots, G_n\}$ coming from $k$ classes, we first compute the unweighted persistence images $A_i$ for each graph $G_i$, and apply the framework from Section 3.1 to learn the "best" weight-function $\omega^* : \mathbb{R}^2 \to \mathbb{R}$ on the birth-death plane $\mathbb{R}^2$ using these persistence images $\{A_1, \ldots, A_n\}$ and their labels. We then perform graph classification using kernel-SVM with the learned $\omega^*$-WKPI kernel. We refer to this framework as *WKPI-classification* framework. We show two sets of experiments. Section 4.1 shows that our learned WKPI kernel significantly outperforms existing persistence-based representations. In Section 4.2, we compare the performance of WKPI-classification framework with various state-of-the-art methods for the graph classification task over a range of data sets. More details / results can be found in Supplement Section 2.

**Setup for our WKPI-based framework.**   In all our experiments, we assume that the weight-function comes from the class $\mathcal{F}$ of mixture of $m$ 2D non-negatively weighted Gaussians as described in the end of Section 3.2. We take $m$ and the width $\sigma$ in our WKPI kernel as hyperparameters: Specifically, we search among $m \in \{3, 4, 5, 6, 7, 8\}$ and $\sigma \in \{0.001, 0.01, 0.1, 1, 10, 100\}$. The $10 * 10$-fold nested cross validation are applied to evaluate our algorithm: There are 10 folds in outer loop for evaluation of the model with selected hyperparameters and 10 folds in inner loop for hyperparameter tuning. We then repeat this process 10 times (although the results are extremely close whether repeating 10 times or not). Our optimization procedure terminates when the change of the cost function remains $\leq 10^{-4}$ or the iteration number exceeds 2000.

Table 1: Classification accuracy on neuron dataset. Our results are WKPI-km and WKPI-kc.

| Datasets | Existing approaches | | | Alternative metric learning | | Our WKPI framework | |
|---|---|---|---|---|---|---|---|
| | PWGK | SW | PI-PL | altWKPI | trainPWGK | WKPI-km | WKPI-kc |
| NEURON-BINARY | 80.5±0.4 | 85.3±0.7 | 83.7±0.3 | 82.1±2.1 | 84.6±2.4 | **89.6 ±2.2** | 86.4±2.4 |
| NEURON-MULTI | 45.1±0.3 | 57.6±0.6 | 44.2±0.3 | 54.3±2.3 | 49.7±2.4 | 56.6±2.7 | **59.3±2.3** |
| Average | 62.80 | 71.45 | 63.95 | 68.20 | 67.15 | **73.10** | 72.85 |

One important question is to initialize the centers of the Gaussians in our mixture. There are three strategies that we consider. (1) We simply sample $m$ centers in the domain of persistence images randomly. (2) We collect all points in the persistence diagrams $\{A_1, \ldots, A_n\}$ derived from the training data $\Xi$, and perform a k-means algorithm to identify $m$ means. (3) We perform a k-center algorithm to those points to identify $m$ centers. Strategies (2) and (3) usually outperform strategy (1). Thus in what follows we only report results from using k-means and k-centers as initialization, referred to as *WKPI-kM* and *WKPI-kC*, respectively.

## 4.1 Comparison with other persistence-based methods

We compare our methods with state-of-the-art persistence-based representations, including the Persistence Weighted Gaussian Kernel (PWGK) [33], original Persistence Image (PI) [1], and Sliced Wasserstein (SW) Kernel [12]. Furthermore, as mentioned in *Remark 2* after Definition 3.1, we can learn weight functions in PWGK by the optimizing the same cost function (via replacing our WKPI-distance with the one computed from PWGK kernel); and we refer to this as trainPWGK. We can also use an alternative kernel for persistence images as described in *Remark 2*, and then optimize the same cost function using distance computed from this kernel; we refer to this as altWKPI. We will compare our methods both with existing approaches, as well as with these two alternative metric-learning approaches (trainPWGK and altWKPI).

**Generation of persistence diagrams.** Neuron cells have natural tree morphology, rooted at the cell body (soma), with dendrite and axon branching out, and are commonly modeled as geometric trees. See Figure 1 in the Supplement for an example. Given a neuron tree $T$, following [36], we use the descriptor function $f : T \to \mathbb{R}$ where $f(x)$ is the geodesic distance from $x$ to the root of $T$ along the tree. To differentiate the dendrite and axon part of a neuron cell, we further negate the function value if a point $x$ is in the dendrite. We then use the union of persistence diagrams $A_T$ induced by both the sublevel-set and superlevel-set filtrations w.r.t. $f$. Under these filtrations, intuitively, each point $(b, d)$ in the birth-death plane $\mathbb{R}^2$ corresponds to the creation and death of certain branch feature for the input neuron tree. The set of persistence diagrams obtained this way (one for each neuron tree) is the input to our WKPI-classification framework.

**Results on neuron datasets.** **Neuron-Binary** dataset consists of 1126 neuron trees from two classes; while **Neuron-Multi** contains 459 neurons from four classes. As the number of trees is not large, we use all training data to compute the gradients in the optimization process instead of mini-batch sampling. Persistence images are both needed for the methodology of [1] and as input for our WKPI-distance, and its resolution is fixed at roughly $40 \times 40$ (see Supplement 2.2 for details). For persistence image (PI) approach of [1], we experimented both with the unweighted persistence images (PI-CONST), and one, denoted by (PI-PL), where the weight function $\alpha : \mathbb{R}^2 \to \mathbb{R}$ is a simple piecewise-linear (PL) function adapted from what's proposed in [1]; see Supplement 2.2 for details. Since PI-PL performs better than PI-CONST on both datasets, Table 1 only shows the results of PI-PL. The classification accuracy of various methods is given in Table 1. Our results are consistently better than other topology-based approaches, as well as alternative metric-learning approaches; not only for the neuron datasets as in Table 1, but also for graph benchmark datasets shown in Table 3 of Supplement Section 2.2, and often by a large margin. In Supplement Section 2.1, we also show the heatmaps indicating the learned weight function $\omega : \mathbb{R}^2 \to \mathbb{R}$.

## 4.2 Graph classification task

We use a range of benchmark datasets: (1) several datasets on graphs derived from small chemical compounds or protein molecules: **NCI1** and **NCI109** [44], **PTC** [24], **PROTEIN** [6], **DD** [21] and **MUTAG** [19]; (2) two datasets on graphs representing the response relations between users in Reddit: **REDDIT-5K** (5 classes) and **REDDIT-12K** (11 classes) [48]; and (3) two datasets on

Table 2: Graph classification accuracy + standard deviation. Our results are last two columns.

| Dataset | Previous approaches | | | | | | | Our approaches | |
|---|---|---|---|---|---|---|---|---|---|
| | RetGK | WL | DGK | P-WL-UC | PF | PSCN | GIN | WKPI-kM | WKPI-kC |
| NCI1 | 84.5±0.2 | 85.4±0.3 | 80.3±0.5 | 85.6±0.3 | 81.7±0.8 | 76.3±1.7 | 82.7±1.6 | 87.5±0.5 | 84.5±0.5 |
| NCI109 | - | 84.5±0.2 | 80.3±0.3 | 85.1±0.3 | 78.5±0.5 | - | - | 85.9±0.4 | 87.4±0.3 |
| PTC | 62.5±1.6 | 55.4±1.5 | 60.1±2.5 | 63.5±1.6 | 62.4±1.8 | 62.3±5.7 | 66.6±6.9 | 62.7±2.7 | 68.1±2.4 |
| PROTEIN | 75.8±0.6 | 71.2±0.8 | 75.7±0.5 | 75.9±0.8 | 75.2±2.1 | 75.0±2.5 | 76.2±2.6 | 78.5±0.4 | 75.2±0.4 |
| DD | 81.6±0.3 | 78.6±0.4 | - | 78.5±0.4 | 79.4±0.8 | 76.2±2.6 | - | 82.0±0.5 | 80.3±0.4 |
| MUTAG | 90.3±1.1 | 84.4±1.5 | 87.4±2.7 | 85.2±0.3 | 85.6±1.7 | 89.0±4.4 | 90.0±8.8 | 85.8±2.5 | 88.3±2.6 |
| IMDB-BINARY | 71.9±1.0 | 70.8±0.5 | 67.0±0.6 | 73.0±1.0 | 71.2±1.0 | 71.0±2.3 | 75.1±5.1 | 70.7±1.1 | 75.1±1.1 |
| IMDB-MULTI | 47.7±0.3 | 49.8±0.5 | 44.6±0.4 | - | 48.6±0.7 | 45.2±2.8 | 52.3±2.8 | 46.4±0.5 | 49.5±0.4 |
| REDDIT-5K | 56.1±0.5 | 51.2±0.3 | 41.3±0.2 | - | 56.2±1.1 | 49.1±0.7 | 57.5±1.5 | 59.1±0.5 | 59.5±0.6 |
| REDDIT-12K | 48.7±0.2 | 32.6±0.3 | 32.2±0.1 | - | 47.6±0.5 | 41.3±0.4 | - | 47.4±0.6 | 48.4±0.5 |

IMDB networks of actors/actresses: **IMDB-BINARY** (2 classes), and **IMDB-MULTI** (3 classes). See Supplement Section 2.2 for descriptions of these datasets, and their statistics (sizes of graphs etc).

Many graph classification methods have been proposed in the literature, with different methods performing better on different datasets. Thus we include multiple approaches to compare with, to include state-of-the-art results on different datasets: six graph-kernel based approaches: RetGK[50], Weisfeiler-Lehman kernel (WL)[44], Weisfeiler-Lehman optimal assignment kernel (WL-OA)[32], Deep Graphlet kernel (DGK)[48], the very recent persistent Weisfeiler-Lehman kernel (P-WL-UC) [42], and Persistence Fisher kernel[34]; two graph neural networks: PATCHYSAN (PSCN) [39] and Graph Isomorphism Network (GIN)[46].

**Classification results.** To generate persistence summaries, we need a meaningful descriptor function on input graphs. We consider two choices: (a) the *Ricci-curvature function* $f_c : G \to \mathbb{R}$, where $f_c(x)$ is the discrete Ricci curvature for graphs as introduced in [37]; and (b) *Jaccard-index function* $f_J : G \to \mathbb{R}$ which measures edge similarities in a graph. See Supplement 2.2 for details. Graph classification results are in Table 2: Ricci curvature function is used for the small chemical compounds datasets (NCI1, NCI9, PTC and MUTAG), while Jaccard function is used for proteins datasets (PROTEIN and DD) and the social/IMDB networks (IMDB's and REDDIT's). Results of previous methods are taken from their respective papers. Comparisons with **more methods** (including with other topology-based methods such as SW [12]) are in Supplement Section 2.2. We rerun the two best performing approaches GIN and RetGK using the exactly same nested cross validation setup as ours. The results are also in Supplement Section 2.2, which are similar to those in Table 2.

Except for **MUTAG** and **IMDB-MULTI**, the performances of our WKPI-framework are similar or better than **the best of other methods**. Our WKPI-framework performs well on both chemical graphs and social graphs, while some of the earlier work tend to work well on one type of the graphs. Furthermore, note that the chemical / molecular graphs usually have attributes associated with them. Some existing methods use these attributes in their classification [48, 39, 50]. Our results however are obtained **purely based on graph structure** without using any attributes. In terms of variance, the standard deviations of our methods tend to be on-par with graph kernel based previous approaches; and are usually much better (smaller) than the GNN based approaches (i.e, PSCN and GIN).

## 5 Concluding remarks

This paper introduces a new weighted-kernel for persistence images (WKPI), together with a metric-learning framework to learn the best weight-function for WKPI-kernel from labelled data. We apply the learned WKPI-kernel to the task of graph classification, and show that our new framework achieves similar or better results than the best results among a range of previous approaches.

In our current framework, only a single descriptor function of each input object is used to derive a persistence-based representation. It will be interesting to extend our framework to leverage multiple descriptor functions (so as to capture different types of information) effectively. Recent work on multidimensional persistence would be useful in this effort. Another interesting question is to study how to incorporate categorical attributes associated to graph nodes effectively. Real-valued attributed can be used as a descriptor function to generate persistence-based summaries. But the handling of categorical attributes via topological summary is much more challenging, especially when there is no (prior-known) correlation between these attributes (e.g, the attribute is simply a number from $\{1, 2, \cdots, s\}$, coming from $s$ categories. The indices of these categories may carry no meaning).

**Acknowledgments**

The authors would like to thank Chao Chen and Justin Eldridge for useful discussions related to this project. We would also like to thank Giorgio Ascoli for helping provide the neuron dataset. This work is partially supported by National Science Foundation via grants CCF-1740761, CCF-1733798, and RI-1815697, as well as by National Health Institute under grant R01EB022899.

## Footnotes

[1]In fact, we can define our kernel without transforming the persistence diagram. We use the transformation simply to follow the same convention as persistence images.

[2] In our implementation, we add a penalty term $\sum_{r=1}^{m} \frac{c}{exp(w_r)}$ to total-cost $k - Tr(HLH^T)$, to achieve this in a "soft" manner.

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
