[Supplementary Material]

# ID 5218: Supplementary Materials

## 1 Missing details from Section 3

### 1.1 Proof of Lemma 3.2

Consider an arbitrary collection of $n$ persistence images $\{\mathrm{PI}_1, \ldots, \mathrm{PI}_n\}$ (i.e, a collection of $n$ vectors in $\mathbb{R}^N$). Set $K = [k_{ij}]_{n \times n}$ to be the $n \times n$ kernel matrix where $k_{ij} = k_w(\mathrm{PI}_i, \mathrm{PI}_j)$. Now given any vector $v = (v_1, v_2, ..., v_n)^T$, we have that:

$$
\begin{aligned}
v^T K v &= \sum_{i,j=1}^{n} v_i v_j k_{ij} \\
&= \sum_{i,j=1}^{n} v_i v_j \sum_{s=1}^{m} \omega(p_s) e^{-\frac{(\mathrm{PI}_i(s) - \mathrm{PI}_j(s))^2}{2\sigma^2}} \\
&= \sum_{s=1}^{m} \omega(p_s) \sum_{i,j=1}^{n} v_i v_j e^{-\frac{(\mathrm{PI}_i(s) - \mathrm{PI}_j(s))^2}{2\sigma^2}}.
\end{aligned}
$$

Because Gaussian kernel is positive semi-definite and the weight-function $\omega$ is non-negative, $v^T K v \geq 0$ for any $v \in \mathbb{R}^N$. Hence the WKPI kernel is positive semi-definite.

### 1.2 Proof of Theorem 3.4

By Definitions **3.1** and **3.3**, combined with the fact that $1 - e^{-x} \leq x$ for any $x \in \mathbb{R}$, we have that:

$$
\begin{aligned}
\mathrm{D}_\omega{}^2(A, B) &= k_w(\mathrm{PI}_A, \mathrm{PI}_A) + k_w(\mathrm{PI}_B, \mathrm{PI}_B) - 2k_w(\mathrm{PI}_A, \mathrm{PI}_B) \\
&= 2\sum_{s=1}^{N} \omega(p_s) - 2\sum_{s=1}^{n} \omega(p_s) e^{-\frac{(\mathrm{PI}_A(s) - \mathrm{PI}_B(s))^2}{\sigma^2}} \\
&= 2\sum_{s=1}^{N} \omega(p_s)(1 - e^{-\frac{(\mathrm{PI}_A(s) - \mathrm{PI}_B(s))^2}{\sigma^2}}) \\
&\leq 2c_w \sum_{s=1}^{N} (1 - e^{-\frac{(\mathrm{PI}_A(s) - \mathrm{PI}_B(s))^2}{\sigma^2}}) \\
&\leq 2\frac{c_w}{\sigma^2} \sum_{s=1}^{n} (\mathrm{PI}_A(s) - \mathrm{PI}_B(s))^2 \\
&\leq 2\frac{c_w}{\sigma^2} \|\mathrm{PI}_A - \mathrm{PI}_B\|_2^2
\end{aligned}
$$

Furthermore, by Theorem 10 of [1], when the distribution $\phi_u$ to in Definition 2.1 is the normalized Gaussian $\phi_u(z) = \frac{1}{2\pi\tau^2} e^{-\frac{\|z-u\|^2}{2\tau^2}}$, and the weight function $\alpha = 1$, we have that $\|\mathrm{PI}_A - \mathrm{PI}_B\|_2 \leq \sqrt{\frac{10}{\pi}} \cdot \frac{1}{\tau} \cdot d_{W,1}(A, B)$. (Intuitively, view two persistence diagrams $A$ and $B$ as two (appropriate)

measures, and $d_{W,1}(A, B)$ is then the "earth-mover" distance between them so as to convert the measure corresponding to $A$ to that for $B$, where the cost is measured by the total $L_1$-distance that all mass have to travel.) Combining this with the inequalities for $D_\omega{}^2(A, B)$ above, the theorem then follows.

## 1.3   Proof of Theorem 3.6

We first show the following properties of matrix $L$ which will be useful for the proof later.

**Lemma 1.1** *The matrix L is symmetric and positive semi-definite. Furthermore, for every vector $f \in \mathbb{R}^n$, we have*

$$f^T L f = \frac{1}{2} \sum_{i,j=1}^{n} \Lambda_{ij}(f_i - f_j)^2 \tag{1}$$

*Proof:* By construction, it is easy to see that $L$ is symmetric as matrices $\Lambda$ and $G$ are. The positive semi-definiteness follows from Eqn (1) which we prove now.

$$
\begin{aligned}
f^T L f = f^T G f - f^T \Lambda f \ &= \sum_{i=1}^{n} f_i^2 g_{ii} - \sum_{i,j=1}^{n} f_i f_j \Lambda_{ij} \\
&= \frac{1}{2} \Big( \sum_{i=1}^{n} f_i^2 g_{ii} + \sum_{j=1}^{n} f_j^2 g_{jj} - \sum_{i,j=1}^{n} 2 f_i f_j \Lambda_{ij} \Big) \\
&= \frac{1}{2} \Big( \sum_{i=1}^{n} f_i^2 \sum_{j=1}^{n} \Lambda_{ij} + \sum_{j=1}^{n} f_j^2 \sum_{i=1}^{n} \Lambda_{ji} \\
&\qquad - \sum_{i,j=1}^{n} 2 f_i f_j \Lambda_{ij} \Big) \\
&= \frac{1}{2} \sum_{i,j=1}^{n} \Lambda_{ij} \cdot (f_i^2 + f_j^2 - 2 f_i f_j) \\
&= \frac{1}{2} \sum_{i,j=1}^{n} \Lambda_{ij}(f_i - f_j)^2
\end{aligned}
$$

The lemma then follows. ∎

We now prove the statement in Theorem 3.6. Recall that the definition of various matrices, and that $h_t$'s are the row vectors of matrix $H$. For simplicity, in the derivations below, we use $D(i, j)$ to denote the $\omega$-induced WKPI-distance $D_\omega(A_i, A_j)$ between persistence diagrams $A_i$ and $A_j$. Applying Lemma 1.1, we have:

$$
\begin{aligned}
\operatorname{Tr}(HLH^T) = \sum_{t=1}^{k}(HLH^T)_{tt} \ &= \sum_{t=1}^{k} h_t L h_t^T \\
&= \sum_{t=1}^{k} \frac{1}{2} \cdot \sum_{j_1,j_2=1}^{n} D^2(j_1, j_2)(h_{t,j_1} - h_{t,j_2})^2 \tag{2} \\
&= \sum_{t=1}^{k} \frac{1}{2} \cdot \sum_{j_1,j_2=1}^{n} D^2(j_1, j_2)(h_{t,j_1}^2 + h_{t,j_2}^2 - 2 h_{t,j_1} h_{t,j_2}).
\end{aligned}
$$

Now by definition of $h_{ti}$, it is non-zero only when $i \in \mathcal{C}_t$. Combined with Eqn (2), it then follows that:

$$
\begin{aligned}
\mathrm{Tr}(HLH^T) &= \sum_{t=1}^{k} \frac{1}{2} \cdot \Big( \sum_{j_1 \in \mathcal{C}_t, j_2 \in [1,n]} \frac{D^2(j_1, j_2)}{cost_\omega(t, \cdot)} \\
&\quad + \sum_{j_1 \in [1,n], j_2 \in \mathcal{C}_t} \frac{D^2(j_1, j_2)}{cost_\omega(t, \cdot)} - 2 \sum_{j_1, j_2 \in \mathcal{C}_t} \frac{D^2(j_1, j_2)}{cost_\omega(t, \cdot)} \Big) \\
&= \sum_{t=1}^{k} \frac{1}{2} \Big( \sum_{j_1 \in \mathcal{C}_t, j_2 \notin \mathcal{C}_t} \frac{D^2(j_1, j_2)}{cost_\omega(t, \cdot)} \\
&\quad + \sum_{j_1 \notin \mathcal{C}_t, j_2 \in \mathcal{C}_t} \frac{D^2(j_1, j_2)}{cost_\omega(t, \cdot)} \Big) \\
&= \sum_{t=1}^{k} \sum_{j_1 \in A_t, j_2 \notin A_t} \frac{D^2(j_1, j_2)}{cost_\omega(t, \cdot)} \\
&= \sum_{t=1}^{k} \frac{cost_\omega(t, \cdot) - cost_\omega(t, t)}{cost_\omega(t, \cdot)} \\
&= k - TC(\omega)
\end{aligned}
$$

This proves the first statement in Theorem 3.6. We now show that the matrix $HGH^T$ is the $k \times k$ identity matrix $\mathbf{I}$. Specifically, first consider $s \neq t \in [1, k]$; we claim:

$$
(HGH^T)_{st} = h_s G h_t^T = \sum_{j_1, j_2 = 1}^{n} h_{sj_1} G_{j_1 j_2} h_{tj_2} = 0.
$$

It equals to 0 because $h_{sj_1}$ is non-zero only for $j_1 \in \mathcal{C}_s$, while $h_{tj_2}$ is non-zero only for $j_2 \in \mathcal{C}_t$. However, for such a pair of $j_1$ and $j_2$, obviously $j_1 \neq j_2$, which means that $G_{j_1 j_2} = 0$. Hence the sum is 0 for all possible $j_1$ and $j_2$'s.

Now for the diagonal entries of the matrix $HGH^T$, we have that for any $t \in [1, k]$:

$$
\begin{aligned}
(HGH^T)_{tt} = h_t G h_t^T &= \sum_{j_1, j_2 = 1}^{n} h_{tj_1} G_{j_1, j_2} h_{tj_2} \\
&= \sum_{j_1, j_2 \in \mathcal{C}_t} \frac{G_{j_1 j_2}}{cost_\omega(t, \cdot)} = \sum_{j_1 \in \mathcal{C}_t} \frac{G_{j_1 j_1}}{cost_\omega(t, \cdot)} \\
&= \sum_{j_1 \in \mathcal{C}_t} \frac{\sum_{\ell=1}^{n} D^2(j_1, \ell)}{cost_\omega(t, \cdot)} \\
&= \frac{\sum_{j_1 \in \mathcal{C}_t, \ell \in [1,n]} D^2(j_1, \ell)}{cost_\omega(t, \cdot)} \\
&= \frac{cost_\omega(t, \cdot)}{cost_\omega(t, \cdot)} = 1.
\end{aligned}
$$

This finishes the proof that $HGH^T = \mathbf{I}$, and completes the proof of Theorem 3.6.

## 2 More details for Experiments

### 2.1 More on neuron experiments

**Description of neuron datasets.** Neuron cells have natural tree morphology (see Figure 1 (a) for an example), rooted at the cell body (soma), with dentrite and axon branching out. Furthermore, this tree morphology is important in understanding neurons. Hence it is common in the field of

neuronscience to model a neuron as a (geometric) tree (see Figure 1 (b) for an example downloaded from NeuroMorpho.Org[2]).

Our NEURON-BINARY dataset consists of 1126 neuron trees classified into two (primary) classes: *interneuron* and *principal neurons* (data partly from the Blue Brain Project [15] and downloaded from http://neuromorpho.org/). The second NEURON-MULTI dataset is a refinement of the 459 interneuron class into four (secondary) classes: *basket-large, basket-nest, neuglia* and *martino*.

(a)                    (b)

Figure 1: (a) An neuron cell (downloaded from Wikipedia)and (b) an example of a neuron tree (downloaded from NeuroMorpho.Org).

**Setup for persistence images.**    Persistence-images are both needed for the methodology of [1] and as input for our WKPI-distance. For each dataset, the persistence image for each object inside is computed within the rectangular bounding box of the points from all persistence diagrams of input trees. The $y$-direction is then discretized to $40$ uniform intervals, while the $x$-direction is discretized accordingly so that each pixel is a square. For persistence image (PI) approach of [1], we show results both for the unweighted persistence images (PI-CONST), and one, denoted by PI-PL, where the weight function $\alpha : \mathbb{R}^2 \to \mathbb{R}$ (for Definition 2.1) is the following piecewise-linear function (modified from one proposed by Adams et al. [1]) where $b$ the largest persistence for any persistent-point among all persistence diagrams.

$$\alpha(x,y) = \begin{cases} \frac{|y-x|}{b} & |y - x| < b \text{ and } y > 0 \\ \frac{|-y-x|}{b} & |-y-x| < b \text{ and } y < 0 \\ 1 & otherwise \end{cases} \tag{3}$$

**Weight function learnt.**    In Figure 2 we show the heatmaps of the learned weight-function $\omega^*$ for both datasets. Interestingly, we note that the important branching features (points in the birth-death plane with high $\omega^*$ values) separating the two primary classes (i.e, for **Neuron-Binary** dataset) is different from those important for classifying neurons from one of the two primary classes (the interneuron class) into the four secondary classes (i.e, the **Neuron-Multi** dataset). Also high importance (weight) points may not have high persistence. In the future, it would be interesting to investigate whether the important branch features are also biochemically important.

Figure 2: Heatmaps of the learned weight-function $\omega^*$ for Neuron-Binary (left) and Neuron-Multi (right) datasets. Each point in this plane indicates the birth-death of some branching feature. Warmer color (e.g, red) indicates higher $\omega^*$ value. $x$- and $y$-axies are birth / death time measured by the descriptor function $f$ (modified geodesic function, where for points in dendrites they are negation of the distance to root).

## 2.2 More on graph classification experiments

**Benchmark datasets for graph classification.**  Below we first give a brief description of the benchmark datasets we used in our experiments. These are collected from the literature.

**NCI1** and **NCI109** [18] consist of two balanced subsets of datasets of chemical compounds screened for activity against non-small cell lung cancer and ovarian cancer cell lines, respectively.
**PTC** [8] is a dataset of graph structures of chemical molecules from rats and mice which is designed for the predictive toxicology challenge 2000-2001.
**DD** [7] is a data set of 1178 protein structures. Each protein is represented by a graph, in which the nodes are amino acids and two nodes are connected by an edge if they are less than 6 Angstroms apart. They are classified according to whether they are enzymes or not.
**PROTEINS** [3] contains graphs of protein. In each graph, a node represents a secondary structure element (SSE) within protein structure, i.e. helices, sheets and turns. Edges connect nodes if they are neighbours along amino acid sequence or neighbours in protein structure space. Every node is connected to its three nearest spatial neighbours.
**MUTAG** [6] is a dataset collecting 188 mutagenic aromatic and heteroaromatic nitro compounds labelled according to whether they have a mutagenic effect on the Gramnegtive bacterium Salmonella typhimurium.
**REDDIT-5K** and **REDDIT-12K** [21] consist of graph representing the discussions on the online forum Reddit. In these datasets, nodes represent users and edges between two nodes represent whether one of these two users leave comments to the other or not. In REDDIT-5K, graphs are collected from 5 sub-forums, and they are labelled by to which sub-forums they belong. In REDDIT-12K, there are 11 sub-forums involved, and the labels are similar to those in REDDIT-5K.
**IMDB-BINARY** and **IMDB-MULTI** [21] are dataset consists of networks of 1000 actors or actresses who played roles in movies in IMDB. In each graph, a node represents an actor or actress, and an edge connects two nodes when they appear in the same movie. In IMDB-BINARY, graphs are classified into Action and Romance genres. In IMDB-MULTI, they are collected from three different genres: Comedy, Romance and Sci-Fi.

The statistics of these datasets are provided in Table 1. In our experiments, for REDDIT-12K dataset, due to the larger size of the dataset (with about 13K graphs), we deploy the EigenPro method ([14], code available at https://github.com/EigenPro/EigenPro-matlab), which is a preconditioned (stochastic) gradient descent iteration) to significantly improve the efficiency of kernel-SVM.

Table 1: Statistics of the benchmark graph datasets

| Dataset | #classes | #graphs | average #nodes | average #edges |
|---|---|---|---|---|
| NCI1 | 2 | 4110 | 29.87 | 32.30 |
| NCI109 | 2 | 4127 | 29.68 | 31.96 |
| PTC | 2 | 344 | 14.29 | 14.69 |
| PROTEIN | 2 | 1113 | 39.06 | 72.82 |
| DD | 2 | 1178 | 284.32 | 715.66 |
| IMDB-BINARY | 2 | 1000 | 19.77 | 96.53 |
| IMDB-MULTI | 3 | 1500 | 13.00 | 65.94 |
| REDDIT-5K | 5 | 4999 | 508.82 | 594.87 |
| REDDIT-12K | 11 | 12929 | 391.41 | 456.89 |

**Persistence generation.**  To generate persistence diagram summaries, we want to put a meaningful descriptor function on input graphs. We consider two choices in our experiments: (a) the *Ricci-curvature function* $f_c : G \to \mathbb{R}$, where $f_c(x)$ is a discrete Ricci curvature for graphs as introduced in [13]; and (b) *Jaccard-index function* $f_J : G \to \mathbb{R}$.

Then Ollivier's Ricci curvature between two nodes $u$ and $v$ is $\kappa_{uv}^\alpha = 1 - W(m_u^\alpha, m_v^\alpha)/d(u, v)$ where $W(\cdot, \cdot)$ is Wasserstein distance between two measures and $d(u, v)$ is the distance between two nodes,

and probability measure $m_u^\alpha$ around node $u$ is defined as

$$m_x^\alpha(x) = \begin{cases} \alpha & x = u \\ (1-\alpha)/n_u & x \in \mathcal{N}(u) \\ 0 & \text{otherwise} \end{cases} \tag{4}$$

$n_u = |\mathcal{N}(u)|$ and $\alpha$ is a parameter within $[0,1]$. In this paper, we set $\alpha = 0.5$.

In particular, the Jaccard-index of an edge $(u,v) \in G$ in the graph is defined as $\rho(u,v) = \frac{|NN(u) \cap NN(v)|}{|NN(u) \cup NN(v)|}$, where $NN(x)$ refers to the set of neighbors of node $x$ in $G$. The Jaccard index has been commonly used as a way to measure edge-similarity[1]. As in the case for neuron data sets, we take the union of the $0$-th persistence diagrams induced by both the sublevel-set and the superlevel-set filtrations of the descriptor function $f$, and convert it to a persistence image as input to our WKPI-classification framework [2].

In all results reported in main text and in Table 2, Ricci curvature function is used for the small chemical compounds data sets (NCI1, NCI9, PTC and MUTAG), while Jaccard function is used for the two proteins datasets (PROTEIN and DD) as well as the social/IMDB networks (IMDB's and REDDIT's). Both 0-dim and 1-dim extented persistence diagrams are employed. In general, we observe that Ricci curvature is more sensitive to accurate graph local structure, while Jaccard function is better for noisy graphs (with noisy edge). In Figure 3, we show the heatmaps of the weight function before and after our metric learning for NCI1 and REDDIT-5K datasets. In particular, the left column shows the heatmaps of the initialized weight function, while the right column shows the heatmaps of the optimal weight function as learned by our algorithm.

Table 2: Classification accuracy on graphs. Our results are in columns WKPI-kM and WKPI-kC.

| Dataset | Previous approaches | | | | | | | | Our appraches | |
|---|---|---|---|---|---|---|---|---|---|---|
| | RetGK | WL | WL-OA | DGK | FGSD | PSCN | GIN | P-WL-UC | WKPI-kM | WKPI-kC |
| NCI1 | 84.5 | 85.4 | 86.1 | 80.3 | 79.8 | 76.3 | 82.7 | 85.6 | **87.5** | 84.5 |
| NCI109 | - | 84.5 | 86.3 | 80.3 | 78.8 | - | - | 85.1 | 85.9 | **87.4** |
| PTC | 62.5 | 55.4 | 63.6 | 60.1 | 62.8 | 62.3 | 66.6 | 63.5 | 62.7 | **68.1** |
| PROTEIN | 75.8 | 71.2 | 76.4 | 75.7 | 72.4 | 75.0 | 76.2 | 75.9 | **78.5** | 75.2 |
| DD | 81.6 | 78.6 | 79.2 | - | 77.1 | 76.2 | - | 78.5 | **82.0** | 80.3 |
| MUTAG | 90.3 | 84.4 | 84.5 | 87.4 | **92.1** | 89 | 90 | 85.2 | 85.8 | 88.3 |
| IMDB-BINARY | 71.9 | 70.8 | - | 67.0 | 71.0 | 71.0 | 75.1 | 73.0 | 70.7 | **75.4** |
| IMDB-MULTI | 47.7 | 49.8 | - | 44.6 | 45.2 | 45.2 | **52.3** | - | 46.4 | 49.5 |
| REDDIT-5K | 56.1 | 51.2 | - | 41.3 | 47.8 | 49.1 | 57.5 | - | 59.1 | **59.5** |
| REDDIT-12K | **48.7** | 32.6 | - | 32.2 | - | 41.3 | - | - | 47.4 | 48.4 |
| Average | - | 66.39 | - | - | - | - | - | - | 69.99 | **71.66** |

**Additional results.** Many graph classification methods have been proposed in the literature. We compare our results with a range of existing approaches, which includes state-of-the-art results on different datasets: six graph-kernel based approaches: RetGK[22], FGSD[19], Weisfeiler-Lehman kernel (WL)[18], Weisfeiler-Lehman optimal assignment kernel (WL-OA)[10], Deep Graphlet kernel (DGK)[21], and the very recent persistent Weisfeiler-Lehman kernel (P-WL-UC)[3] [17]; two graph neural networks: PATCHYSAN (PSCN) [16], Graph Isomorphism Network (GIN)[20]; as well as the topology-signature-based neural networks [9].

Additional results of comparing our results with more existing methods are given in Table 2. The results of DL-TDA (topological signature based deep learning framework) [9] are not listed in Table 2, as only the classification accuracy for REDDIT-5K (accuracy $54.5\%$) and REDDIT-12K ($44.5\%$) are given in their paper (although their paper contains many more results on other objects, such as images). While also not listed in this table, we note that our results also outperform the newly independently proposed general neural network architecture for persistence representations reported in the very recent preprint [4]. Comparison with other topological-based non-neural network approaches are given below.

Figure 3: Heatmap of initialized weight function (left column) and that of the learnt weight-function $\omega^*$ (right column). Top row shows results for NCI1 data set; while bottom row contains those for REDDIT-5K data set.

**Topological-based methods on graph data.** Here we compare our WKPI-framework with the performance of several state-of-the-art persistence-based classification frameworks, including: PWGK [11], SW [5], PI [1] and PF [12]. We also compare it with two alternative ways to learn the metric for persistence-based representations: **trainPWGK** is the version of PWGK [11] where we learn the weight function in its formulation, using the same cost-function as what we propose in this paper for our WKPI kernel functions. **altWKPI** is the alternative formulation of a kernel for persistence images where we set the kernel to be $k(\mathrm{PI}, \mathrm{PI}') = \sum_{s=1}^{N} e^{-\frac{\omega(p_s)(\mathrm{PI}(s)-\mathrm{PI}'(s))^2}{2\sigma^2}}$, instead of our WKPI-kernel as defined in Definition 3.1.

Table 3: Classification accuracy on graphs for topology-based methods.

| Datasets | Existing TDA approaches | | | | | Alternative metric learning | | Our WKPI framework | |
|---|---|---|---|---|---|---|---|---|---|
| | PWGK | PI-CONST | PI-PL | SW | PF | trainPWGK | altWKPI | deWKPI-kM | deWKPI-kC |
| NCI1 | 73.3 | 72.5 | 72.1 | 80.1 | 81.7 | 76.5 | 77.4 | **87.2** | 84.7 |
| NCI109 | 71.5 | 74.3 | 73.1 | 75.5 | 78.5 | 77.2 | 81.2 | 85.5 | **86.9** |
| PTC | 62.2 | 61.3 | 64.2 | **64.5** | 62.4 | 62.5 | 64.2 | 61.1 | 64.3 |
| PROTEIN | 73.6 | 72.2 | 69.1 | 76.4 | 75.2 | 74.8 | 75.1 | **77.4** | 75.6 |
| DD | 75.2 | 74.2 | 76.8 | 78.9 | 79.4 | 76.4 | 72.5 | **79.8** | 79.1 |
| MUTAG | 82.0 | 85.2 | 83.5 | 87.1 | 85.6 | 86.4 | **88.5** | 85.5 | 88.0 |
| IMDB-BINARY | 66.8 | 65.5 | 69.7 | 69.6 | 71.2 | 71.8 | 67.3 | 70.6 | **75.4** |
| IMDB-MULTI | 43.4 | 42.5 | 46.4 | 48.7 | 48.6 | 45.8 | 45.3 | 47.1 | **48.8** |
| REDDIT-5K | 47.6 | 52.2 | 51.7 | 53.8 | 56.2 | 53.5 | 54.7 | 58.7 | **59.3** |
| REDDIT-12K | 38.5 | 43.3 | 45.7 | **48.3** | 47.6 | 43.7 | 42.1 | 45.2 | 44.5 |
| Average | 63.41 | 64.3 | 65.23 | 68.29 | 68.64 | 66.86 | 66.83 | 69.81 | **70.66** |

Table 4: Graph classification accuracy of GIN and RetGK on graph benchmarks with the same nested cross validation setup

| | NCI1 | NCI109 | PTC | PROTEIN | DD | MUTAG | IMDB-BIN | IMDB-MULTI | Reddit5K | Reddit12K |
|---|---|---|---|---|---|---|---|---|---|---|
| RetGK | 84.5±0.2 | 84.8±0.2 | 62.9±1.6 | 75.4±0.6 | 81.6±0.4 | 90.0±1.1 | 72.3±1.0 | 47.7±0.4 | 55.8 ±0.5 | 48.5± 0.2 |
| GIN | 82.4±1.6 | 86.5±1.5 | 67.8±6.5 | 76.7±2.6 | 81.1±2.5 | 89.0±7.5 | 75.6±5.3 | 52.4±3.1 | 57.2±1.5 | 47.9± 2.1 |

We use the same setup as our WKPI-framework to train these two metrics, and use their resulting kernels for SVM to classify the benchmark graph datasets. WKPI-framework outperforms the existing approaches and alternative metric learning methods on all datasets except **MUTAG**. WKPI-kM (i.e, WKPI-kmeans) and WKPI-kC (i.e, WKPI-kcenter) improve the accuracy by $3.9\% - 11.9\%$ and $5.4\% - 13.5\%$, respectively. Besides, we show results by another experimental setup. In 10-fold cross validation, choose $m$ and $\sigma$ leading to the smallest cost function value, then evaluate the classifier on the test set. Repeat this process 10 times. That is, $m$ and $\sigma$ are not the hyperparameters of the

SVM classifiers, but are determined by the metrics learning. We refer to these two approaches as deWKPI-kM and deWKPI-kC in accordance with the initialization methods. The classification accuracy of all these methods are reported in Table 3.

## Footnotes

[1] We modify our persistence algorithm slightly to handle the edge-valued Jaccard index function

[2] We expect that using the 0-th zigzag persistence diagrams will provide better results. However, we choose to use only 0-th standard persistence as it can be easily implemented to run in $O(n \log n)$ time using a simple union-find data structure.

[3] Note that results for three version of persistent WL kernels are reported in their paper. We take the one (P-WL-UC, with uniform node labels) that performs the best from their Table 1.