[Reviews · NeurIPS 2019]

Reviewer 1



# Update after rebuttal I thank the authors for their detailed rebuttal; my questions were answered sufficiently. I am convinced this would make a good contribution and suggest to accept the paper. # Summary of the review This is a well-written paper on a very relevant topic, namely the use of topological data analysis (TDA) for graph classification. The paper uses a 'learned' weight function for persistence images, i.e. a descriptor of topological features that can be vectorised. The weight function is seen to be crucial for obtaining good performance values. While I have some more comments/suggestions, I suggest acceptance. ## Clarity & terminology The paper is written clearly and can be followed very well. There are a few word choices & explanations that can be improved, though: - Technically, persistence diagrams form a multiset in the _extended_ plane because infinite features might still exist (l. 23). - The caption of Figure 1 could be extended to make the paper more accessible. Space permitting, a brief description of the method, along with pointers to the respective sections/parts of the method could be helpful. - The description of persistent homology in Section 2 is not accessible to non-experts. For example, the sentence 'Using the topological object called homology classes to describe these features' is not entirely correct. I know that the 'typical' description of persistent homology in terms of groups, generators, and so on, is too technical, but I would still suggest rephrasing a few things here. Two suggestions for inspiration (I am not affiliated with any of these authors, I just think that their explanations of persistent homology and TDA are accessible): 1. Carrière et al.: 'PersLay: A Simple and Versatile Neural Network Layer for Persistence Diagrams' 2. Hofer et al.: 'Deep Learning with Topological Signatures' - While correct, I find that the citation of zigzag persistence and extended persistence is not useful here because, as far as I understand, they are not used in the experimental section. - In Figure 2, the persistence image is not rendered correctly; this might be solvable by 'filling' the cells instead of showing only their outlines (it would also make the figure easier to read) - If space permits, some of the equations could be typeset on their own because it would make it easier to refer to them. - In l. 118 (Definition 2.1): what is the 'persistent plane'? This should probably refer to the (extended) persistence plane. - In l 123, there is a sentence fragment: 'The persistence image for a diagram $A$ is $PI_A$ consists of $N$ numbers' - 'Persistence image' is sometimes written as 'persistence-image'. This should be used consistently. - The paper uses $[1, k]$ and $[1, m]$ to denote discrete intervals. I would suggest using a set notation instead; at first glance, the other notation can easily be misunderstood as a continuous interval. - The discussion of the matrix formulation is somewhat redundant: both in l. 196 and l. 223, the paper discusses that this does not require loops. Mentioning this once should be sufficient. - l. 270: This should read 'Generation of persistence diagrams' ## Method & experiments The method and the experiments are well described. The weight function optimisation (and its re-framing as a matrix optimisation algorithm) is described in a straightforward and easy-to-understand manner. The authors are to be commended! A few comments/questions: - Lemma 3.2 could be formulated more generally if I am not mistaken. The proposed method should be a valid kernel for any valid kernel function on the points, right? - How is the training performed exactly? I assume that the hyperparameter optimisation only uses the respective training data set from the current fold? If so, these details should be provided. - What type of filtration is used for the graph classification experiment? A sublevel/superlevel set filtration or extended persistence based on the scalar descriptor? I would expect extended persistence to be used here, because it would explain the additional citations. - The paper discusses additional weight function choices (polynomials of degree $d$) but only Gaussian functions are used for the experiments, unless I am mistaken. Are other functions good enough? If so, this would be an interesting thing to comment on (and maybe even run an additional experiment to empirically demonstrate this). - Is subsampling being used for the experiments or not? The discussion of the complexity leads me to think that subsamples might be employed, but I was unable to find a proper discussion of this. - To be fully self-contained, a definition of the Ricci curvature for graphs could be given (maybe in the supplementary materials). - Heat maps for the neuron trees are being shown (in the supp. mat.); what about the graph classification experiments? The performance in the tables lead me to believe that this might be interesting to show. ## Language & style The paper is written very well. I have only a few highly personal 'pet peeves' or suggestions: - If possible, order citations by their number. The citation string in l. 36 looks a little bit confusing first. - Instead of 'We note that [26]', I personally prefer 'Hofer et al. [26]' because it makes the reference easier to find. Some typos/mistakes: - l. 28: 'Due to these' --> 'Due to these reasons?' - l. 170: 'where $\sigma$ is width' --> 'where $\sigma$ is the width' - l. 204: 'Optimal distance problem' --> 'The optimal distance problem' - Table 2: 'appraches' --> 'approaches' - l. 315: 'Variance speaking' --> 'In terms of variance, ...'

Reviewer 2



The paper proposes a novel kernel for graphs, which is constructed from topological features, namely form persistence diagrams. The major contribution is to learn a weight function for the topological features from labelled graph data, rather than using pre-set fixed weights. In practice, such distance metric is learned on the persistent images, that is a vectorial representation of the persistence diagram. Using the graph labels, the kernel is then specifically optimized to better distinguish between classes. A matrix formulation for speed up calculations is proposed and experiments on graph classification tasks are presented. Both the novel idea and experimental results support the validity of the proposed method. However, I found the technical structure and notation of the paper to be unclear in the model description; the experiments should be made fair by running the competitor methods with the same setup of the proposed approach. Detailed comments are provided below 1) Section 2: the notation and background definitions should be presented more clearly. 2) Concepts such as shape or creation/disruption of topological feature can be introduced in a background section. 3) It would be useful to give practical examples of the topological features to consider for the specific graph application. In particular, the paper should better clarify how the proposed approach is extended to the graph framework. 4) It is unclear for which kind of graphs the method is designed. For instance, how are the features generated from node/edge labels/attributes? 5) The matrix representation idea is strongly emphasized to pursue the argument of efficient computation, mentioning that the optimization is faster in the popular programming language. Could the author(s) provide a more technical explanation? 6) It is mentioned that subsampling techniques for large datasets are used. It would be interesting to discuss how this affects the performances and when it is applied. 7) The manuscript claims statistical significance, but this cannot be guaranteed if the methods are not evaluated on the same splits. Taking the results of the other methods from the papers does make the comparison unfair. %%%%%%%%%%%%%%% Thank you for your response and clarifications. I found the experimental setup to be improved, given the new results that have been provided.

Reviewer 3



--------Update after the rebuttal------ I thank the authors for the rebuttal. However, it seems that some of the concerns have not addressed yet. Firstly, it seems unclear that the optimization problem in 3.2 is formulated as metric learning, but the authors claim that their approach and NCA (even other Mahalanobis metric learning) is fundamentally different (except using the concepts of “in-class” distance or similarity). Indeed, Mahalanobis metric learning learns a linear transformation while the proposed approach learns a weight function. However, a weight function is a special case of a linear transformation when one forces the linear transformation in a diagonal matrix (I understand that you used a weight vector --- a special case of linear transform: diagonal matrix --- over a nonlinear Gaussian mapping). About the non-linear aspect, there is a rich literature about nonlinear metric learning (see the surveys of Bellet et al.’13 and Kulis’12). Therefore, it seems better if the authors can place their proposed approach in the picture of metric learning when claiming the novelty. Currently, I am not sure why the proposed metric learning is novel yet? Secondly, giving a persistence image and using a grid to vectorize it, it is quite surprising that the grid size does not affect the performances of metric learning for classification tasks (following the rebuttal). Somehow, it seems to imply that weights for positions in the persistence image may not be so important in applications (comparing with the uniform weights). Additionally, I see that the authors indeed considered original Persistence Image (PI) as a baseline (line 262). However, it is unclear whether the authors employed Gaussian kernel for Persistence Image (which is a kind of uniform weight for the proposed kernel) or simply apply linear kernel with SVM?). In case, they use linear kernel + PI, I am not sure whether it can give information about the benefits of learning weights, but if they applied Gaussian kernel + PI, then I agree that they illustrate the importance of learning weights. ---> In case, the authors have not considered PI with uniform weight WKPI kernel (or simply Gaussian kernel). It will be better to use this baseline to show whether the learning weight is necessary. (Note that this comment only raised from strange observations from your experimental results in the rebuttal, somehow grid size is not important which seems to imply that uniform weight may be good enough). Somehow, the importance of learning weight may not so convincing through your experiments yet (over the uniform weight). Overall, I feel that although the submission may have some potentials, it still needs more improvements (e.g. above comments, and some of the previous comments such as more care for solving the non-convex optimization problem --- no comments from authors in the rebuttal yet --- it is not sure why the nonconvex optimization is quite easily solved (?), time consumption comparison for experimental results --- partly improved in your rebuttal). However, I appreciate that learning weights is indeed interesting for TDA applications and your empirical results showed some potentials. So, I leave the decision to the area chair. ------------------------------------------ The authors proposed a weighted kernel WKPI for persistence diagrams based on persistence images and a sum of element-wise weighted Gaussian kernel. The authors propose to learn the weight in WKPI as a metric learning problem by using (stochastic) gradient descent (where the weight is corresponding to a node in a grid of persistence images). Therefore, the weight in WKPI is adapted to a given task (learned from training data). Additionally, the authors prove that (i) the proposed kernel is positive definite, which is quite straightforward from its definition, (ii) stability of WKPI-distance w.r.t the 1-Wasserstein by relying on results of persistence images. Empirically, the authors show that the proposed kernel WKPI obtains similar of sometimes significantly better results than other approaches in neuron and graph datasets. The paper is easy to follow. The proposed WKPI obtains good empirical results for neuron and graph datasets. Below are some of my concerns: The authors motivate to learn the weight for points in persistence diagrams (in introduction). However, in the proposed framework, the weight is on the nodes of the grid in persistence image which is a vector-representation for persistence diagrams. It seems that the grid of persistence images is a crucial parameter in experiments. How does it affect the performance in the experiments? The proposed kernel WKPI is based on vector-representation of persistence images for persistence diagrams with L2 metric. There are many metric learning approaches for such kind of problems based on the Mahalanobis distance where one can not only learn the weight for each feature, but also can learn the interaction among these features. In that view, it seems the proposed framework is quite limited. (since the weight on persistence images is quite different to the weight on points in persistence diagrams) The proposed optimization problem for metric learning is quite similar to Neighborhood Component Analysis (NCA) proposed by Goldberger et al.’2005. It seems better if the authors plug the proposed method in a general picture of metric learning and discuss its merit. For the optimization in Section 3.2, it seems it is a nonconvex problem, it seems better if the authors give more care for initialization, local minima, saddle points etc. For the penalty (in footnote page 6), why is it in that form, what is an intuition (since it is quite strange for a regularization.) In experiments, the authors compare WKPI with PWGK and SW in Table 1. It seems that the authors miss Persistence Fisher kernel (Le, Yamada’2018) baseline which is one of state-of-the-art kernels for persistence diagrams. From Table 1, other kernels without learning are quite comparative with the proposed WKPI. It seems better if the authors also illustrate the results of those persistence-kernel on graph data. It is a bit confused why the authors do not consider those kernel approaches as baselines for graph data. It seems better if the authors show the time consumption comparison in the experiments.

[Author Response · NeurIPS 2019]

We thank the reviewers for their comments. We address all major comments below (including clarifying some potential misunderstanding from Reviewer #3). We believe that our paper makes an important first step to develop an effective framework to learn good metrics for persistence summaries; and the performance of our current framework on the challenging graph classification problem is already comparable or better than a range of existing state-of-the-arts approaches in the literature, and also outperforms all previous TDA based methods (sometimes by a large margin). We also note that all code/datasets are already made public and results can be reproduced.

**Response to Reviewer #1.** Thank you for several insightful suggestions (including the generalization of Lemma 3.2). We will incorporate them in the revision. A few clarifications: (1) Indeed: we use 10 times $10 * 10$-fold nested cross validation, and hyperparameter tuning is done in the inner loop using the respective training data from current fold. (2) By "subsampling", we mean the *mini-batch* for stochastic gradient descent during the optimization (for each gradient computation we choose a random subset (mini-batch) of size 50). **All** data from input datasets are used to generate results. For Reddit-12K, as pointed out in Supplement, the EigenPro method by [Ma and Belkin, NIPS 2017] is used to speed up kernel-SVM. But that does not involve subsampling input data. (3) We use extended persistence diagrams for all datasets. (When double-checking results, we noticed that currently what we reported for WKPI-kM for (only) IMDB and Reddit are from using 0-D standard persistence (sub + super levelsets). The results using extended persistence are better: $75.5 \pm 0.1; 51.2 \pm 0.5; 59.5 \pm 0.5; 49.4 \pm 0.6$ for IMDB-Binary, IMDB-Multi, Reddit5K, Reddit12K, respectively.) (4) For heat maps for graph data, we already provide two examples in Figure 3 of the Supplement.

**Response to Reviewer #2.** Thank you for your comments. Your main comment that we should run other methods with the same setup as ours is a very valid point: (i) All results on topology-based methods are already done in exactly the same setup as ours. (ii) We used 10 times 10*10-fold nested cross validation. Most recent work in graph classification literature use (10 times) 10-fold cross validation, which was partly why we didn't re-run the results. We have now re-run the two best performing approaches *GIN* and *RetGK* using our setup. Results are in Table 1: RetGK stays roughly the same. GIN improves slightly, although our results are still comparable or better than it in general. Note that GIN uses **node attributes**, while our results are obtained without them (based purely on graph structure).

Table 1: Accuracy of GIN, RetGK and Persistence Fisher Kernel ($k_{PF}$) on graph benchmark datasets

|  | NCI1 | NCI109 | PTC | PROTEIN | DD | MUTAG | IMDB-BIN | IMDB-MULTI | Reddit5K | Reddit12K |
|---|---|---|---|---|---|---|---|---|---|---|
| RetGK | 84.5±0.2 | 84.8±0.2 | 62.9±1.6 | 75.4±0.6 | 81.6±0.4 | 90.0±1.1 | 72.3±1.0 | 47.7±0.4 | 55.8±0.5 | 48.5±0.2 |
| GIN | 82.4±1.6 | 86.5±1.5 | 67.8±6.5 | 76.7±2.6 | 81.1±2.5 | 89.0±7.5 | 75.6±5.3 | 52.4±3.1 | 57.2±1.5 | 47.9±2.1 |
| $k_{PF}$ | 81.7±0.2 | 78.5±0.3 | 62.4±1.2 | 75.2±0.3 | 79.4±0.3 | 85.6±0.5 | 71.2±0.7 | 48.6±0.2 | 56.2±0.4 | 47.6±0.3 |

We also provide a few clarifications: (1) For "subsampling technique": please see point (2) in our response to Reviewer #1 above. (2) Our method treats graphs as **non-attributed** (see e.g, lines 314-315 of submission), and persistence summaries are generated using the Ricci curvature and Jaccard index descriptor functions as described in lines 300-303 of submission. Using only graph structures, we can already obtain similar or better results than those previous approaches using attributes, and it will be interesting to explore in the future whether using attributes can further improve performance. (3) Persistent homology can be computed for graphs in a standard way by treating functions defined on it as a piecewise-linear function. We will elaborate on all these points in the revision, add more background on persistent homology, and provide an intuitive example of persistence for neuron trees to explain the ideas.

**Response to Reviewer #3.** Thank you for your comments. We address / clarify your major comments: (1) *Comparing with other TDA methods on graph datasets:* Indeed, we **already provide** that in Table 3 of Supplement (also see lines 281-284 of submission), and our method outperforms them in all cases. (Note the change of results for WKPI-kM for IMDB+Reddit in our response to Reviewer #1). Results for Persistent Fisher kernel (PF), run in the same setup as ours, are in Table 1 which we will add to the revised paper. The performance of PF on these data is similar to the SW method (which we already compare with) and our method outperforms PF in all cases (sometimes by a large margin).
(2) *Regarding the paper on NCA by Goldberger et al, and metric learning on Mahalanobis distance:* The similarity with NCA is perhaps superficial, mostly in the sense that both intuitively optimize some total "in-class" distance or similarity (which is common in most metric learning approaches). The differences are fundamental. (2.a) NCA utilizes the probability of correct KNN based classification, while idea of our approach comes from spectral clustering, which leads to different precise formulation of the objective function. (2.b) NCA and other metric learning for Mahalanobis distance learns a *linear transformation* of grid points (coordinates) in persistence image (PI), while our approach learns a weight function $\omega : \mathbb{R}^2 \to \mathbb{R}$ defined on the birth-death plane containing PIs. Also, our pseudo-distance on persistence images involve a non-linear kernel. (2.c) NCA is learning a $N \times N$ matrix $A$, where $N$ is the total number of grid points in one PI. The number of parameter in NCA is roughly $N^2$ (or $dN$ for dimension reduction to $d$-D space). However, in our *parametric formulation*, we only learn the parameters of the target weight function: e.g, for $\omega$ being a mixture of $m$ isotropic Gaussians, the number of parameters is $O(m)$, which can be several orders of magnitude smaller than $N^2$.

(3) *Grid size for persistence image (PI):* We don't think, nor have observed that the discretization of PI has significant effect on performance, which is why we didn't set it as a hyperparameter. (Note that the **same size** (see "Setup for persistence images" on Pg 4, Supplement) is used for **all datasets**.) For example, on PROTEIN: the accuracy for grid sizes $s * s$ with $s = 10; 20; 30; 40; 50; 60$ is 76.7; 78.6; 78.5; 78.5; 76.3; 73.2. We will include these in revision.

[Meta-Review · NeurIPS 2019]

The reviewers all considered the author feedback and discussed the paper thoroughly. Most concerns could thereby be clarified but not all. Overall, I consider the contribution sufficiently interesting and valuable to justify accepting the paper as a poster. -- Minor detail: should k_w be k_omega?